# ConLUX:
# Concept-Based Local Unified Explanations

## Abstract

With the rapid advancements of various machine learning models, there is a significant demand for model-agnostic explanation techniques, which can explain these models across different architectures. Mainstream model-agnostic explanation techniques generate local explanations based on basic features (e.g., words for text models and (super-)pixels for image models). However, these explanations often do not align with the decision-making processes of the target models and end-users, resulting in explanations that are unfaithful and difficult for users to understand. On the other hand, concept-based techniques provide explanations based on high-level features (e.g., topics for text models and objects for image models), but most are model-specific or require additional pre-defined external concept knowledge. To address this limitation, we propose ConLUX, a general framework to provide concept-based local explanations for any machine learning models. Our key insight is that we can automatically extract high-level concepts from large pre-trained models, and uniformly extend existing local model-agnostic techniques to provide unified concept-based explanations. We have instantiated ConLUX on four different types of explanation techniques: LIME, Kernel SHAP, Anchor, and LORE, and applied these techniques to text and image models. Our evaluation results demonstrate that 1) compared to the vanilla versions, ConLUX offers more faithful explanations and makes them more understandable to users, and 2) by offering multiple forms of explanations, ConLUX outperforms state-of-the-art concept-based explanation techniques specifically designed for text and image models, respectively.

## 1 Introduction

As machine learning models become more complex and popular, it has become an emerging topic to explain the rationale behind their decisions. In particular, as the structure of machine learning models diversifies and evolves rapidly, and effective closed-source models (e.g., GPT-4 (Achiam et al., 2023) and Gemini (et al., 2024b)) become more prevalent, model-agnostic explanation techniques show their appeal due to their adaptability to various models and tasks (Wang, 2024). These techniques consider target models as black boxes, so they can explain any machine learning model without requiring any knowledge of the model's internal structure. This paper addresses the challenge of incorporating high-level concepts into local model-agnostic techniques to explain the decision-making processes of various machine learning models, including large language models (LLMs).

To faithfully explain the behavior of machine learning models, it is essential to provide explanations built from language components aligned with the decision process of the target models; to make explanations easy to understand, it is also crucial to provide explanations built from user-friendly language components (Poeta et al., 2023a). Unfortunately, mainstream model-agnostic explanation techniques often fail to meet both requirements, as they provide explanations built from basic features (e.g., words for text models and (super-)pixels for image models) (Ribeiro et al., 2016; Lundberg & Lee, 2017; Ribeiro et al., 2018; Guidotti et al., 2018). In contrast, many concept-based techniques provide explanations based on high-level features (e.g., topics in texts and objects in images) (Poeta et al., 2023a). These techniques either utilize the internal information of the target models like gradients, activations, and attention weights (Zhang et al., 2021b; Yeh et al., 2020; 2019b; Cunningham et al., 2023; Ghorbani et al., 2019b; Crabbé & van der Schaar, 2022; Fel et al., 2023), or pre-defined external knowledge (El Shawi, 2024; Widmer et al., 2022) to build high-level

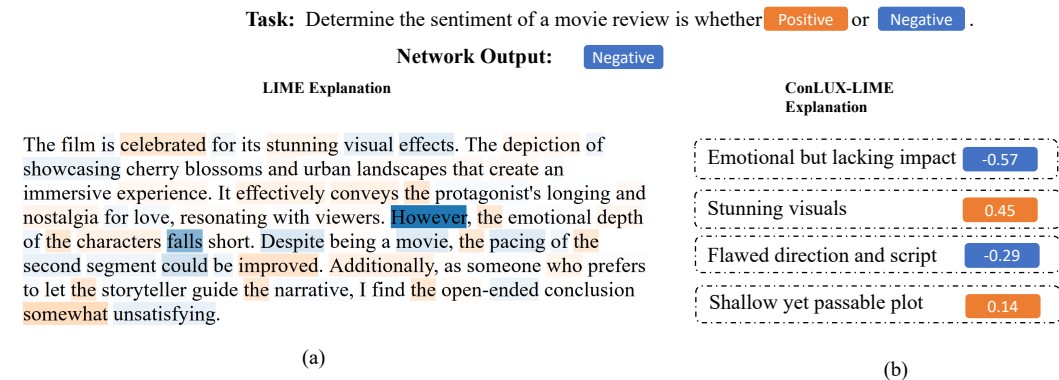

Figure 1: Example explanation (a) is generated by LIME, demonstrating how each word in the input sentence contributes to the target model's prediction. The color intensity reflects the magnitude of the weight, with deeper hues indicating larger absolute values. Example explanation (b) is generated by ConLUX-augmented LIME, providing an explanation based on high-level concepts.

concepts. This limits these techniques to specific types of models or tasks. Furthermore, while there are different forms of explanations (e.g. feature attributions, rules) for various purposes (Zhang et al., 2021c), existing concept-based explanations mainly focus only on attributions, which limits their fidelity and applicability.

To bridge this gap, we aim to elevate the explanations of various forms provided by existing model-agnostic techniques from feature-level to concept-level. As we focus on explaining the decision-making process of machine learning models to end-users, we put our emphasis on local explanations, which are more tractable for end-users. We find that existing local model-agnostic techniques all follow similar workflows, which allows us to introduce a unified way to elevate all these techniques from feature level to concept level. This transition necessitates automating the concept extraction process and establishing bidirectional mappings between concept representations and feature representations for given input data. Noticing that existing works have utilized large pre-trained models to extract concepts and represent input data at the concept level for specific tasks (Ludan et al., 2023; Sun et al., 2023), we generalize these findings and further observe that large models also have the ability to map concept-level representations back to the feature-level. To this end, we propose ConLUX, a general framework that automatically incorporates high-level concepts into various existing local model-agnostic techniques for any machine learning models, and provides local explanations in various forms for diverse user needs.

We take three mainstream local model-agnostic techniques, LIME (Ribeiro et al., 2016), Anchor (Ribeiro et al., 2018), and LORE (Guidotti et al., 2018) as examples to illustrate how ConLUX improves local model-agnostic explanations.

Figure 1 shows a LIME explanation for a BERT-based sentiment analysis model on a movie review. The target model classifies the sentence as negative. LIME provides feature-level attributions, indicating how each word contributes to the model's prediction. In this case, LIME assigns high negative scores to the words "however" and "falls", which indicates that these words contribute much to the negative prediction. However, this explanation is unfaithful and hard to understand by end-users. For example, the word "however" is assigned a high negative score, but it functions as a conjunction and does not directly convey sentiment (Liu & Zhang, 2023). Moreover, such confusing attributions, combined with an overwhelming amount of attribution information, complicate the explanation for end-users. ConLUX addresses these issues by elevating the explanation from feature-level to concept-level. ConLUX-agumented LIME extracts the main topics of the input sentence using GPT-4o, and then provides attribution-based explanations with these topics. From this explanation, users can easily understand that the negative prediction is mainly because the sentence mentions the movie's poor performance in "emotion impact" and "direction and script", while the part about "stunning visuals" and "passable plot" also expresses some positive sentiment. This ex-

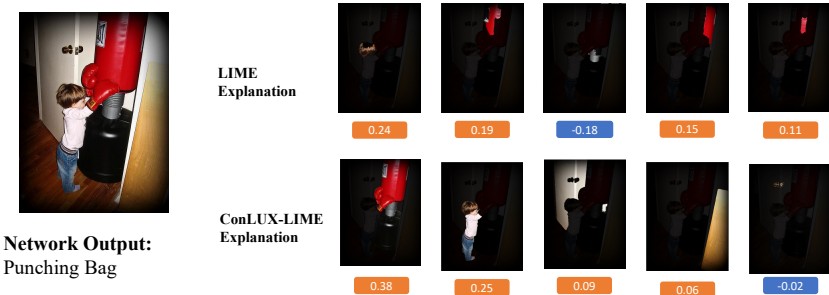

Figure 2: Example explanations generated by LIME (upper) and ConLUX-augmented LIME (lower) for an image classification task.

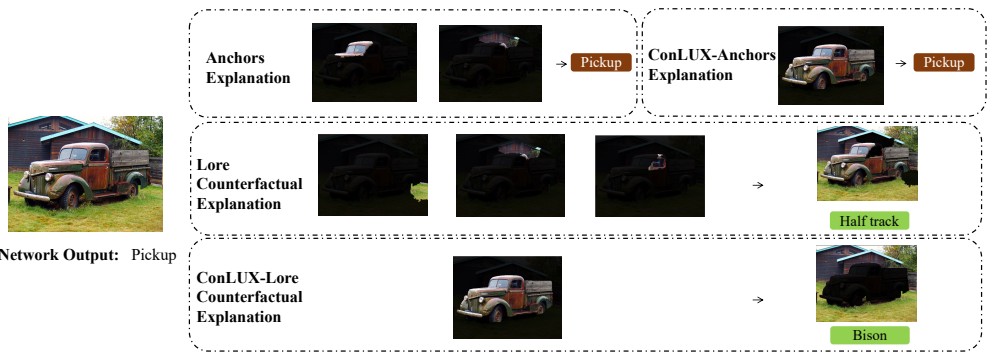

Figure 3: Example Anchors, LORE explanations and their ConLUX-augmented versions.

planation faithfully reflects the decision process of the target model and is more understandable to end-users.

Similar issues exist in the explanation of image models. We use YOLOv8 (Jocher et al., 2023) to perform an image classification task on ImageNet (Deng et al., 2009) dataset. Figure 2 shows a LIME explanation for an image classified as a *punching bag*. LIME attributes high importance to some fragmented superpixels. End-users can hardly understand why these parts are important for the model's prediction. ConLUX-augmented LIME provides explanations based on objects detected by Segment Anything Model (SAM) (Kirillov et al., 2023), and attributes high importance to the punching bag itself and the kid in the image. This explanation is more faithful and understandable. End-users can easily realize that the model does not perform perfectly when classifying this image to a *punching bag*.

Figure 3 shows the explanations generated by Anchor, LORE, and their ConLUX-augmented versions for an image classified as a *pickup*. Anchors provides rule-based sufficient conditions (referred to as *anchors*) for the target model's prediction. The vanilla anchor indicates that parts of the car and the background house guarantee the prediction as a *pickup*. With ConLUX, end-users can easily understand that the model classifies the image as a *pickup* exactly because it indeed detects the pickup truck in the image. LORE provides rule-based sufficient conditions and counterfactual explanations. Figure 3 shows the counterfactual explanations, which show users how to change the model's prediction by modifying the input image. The vanilla LORE explanation indicates that if we mask a part of the grass, the background house and the whole side window of the truck will change the model's prediction to a *Half track*. In contrast, ConLUX-augmented LORE indicates that users can simply mask the pickup truck to change the model's prediction to a *Bison*, which is more user-friendly.

The preceding examples indicate that feature-level explanations are hard to understand by end-users. On the other hand, as high-level concepts align with the decision process of target models and users better (Zhang et al., 2021a; Ghorbani et al., 2019a; Kim et al., 2018; Sun et al., 2023), ConLUX addresses this by providing concept-level explanations, and the examples demonstrate that ConLUX

makes explanations more understandable to end-users. Moreover, our empirical evaluation shows these concept-level explanations are also more faithful to the models. Finally, benefiting from the various types of explanations provided by existing local model-agnostic techniques, ConLUX can provide rich explanations including attributions, sufficient conditions, and counterfactuals, satisfying diverse user needs and offering a more comprehensive understanding of the target models. This fills the current gap in concept-based explanations, which lack forms beyond attributions.

To elevate the explanations provided by existing local model-agnostic techniques from feature-level to concept-level, we modify these techniques in a uniform and lightweight way based on their two commonalities: 1) these techniques use basic features as language components to build explanations; 2) these techniques use a perturbation model, which generates samples similar to the given input by changing some of its feature values, to capture the local behavior of the target model at the feature level. To this end, by only elevating the language components to high-level concepts and extending the perturbation model to generate samples by changing high-level concepts, ConLUX extends these techniques to provide concept-level explanations.

We evaluated ConLUX on explaining two sentiment analysis models (BERT, Llama 3.1(et al., 2024a)) and three image classification models(YOLOv8, Vision Transformer (Oquab et al., 2023; Darcet et al., 2023), and ResNet-50 (He et al., 2016)). Our evaluation results demonstrate that ConLUX improves the fidelity of Anchors, LIME, LORE, and Kernel SHAP explanations by 82.21%, 48.59%, 149.93%, and 48.27% respectively, and by considering various forms of explanations together, ConLUX outperforms state-of-the-art concept-based explanation techniques specifically designed for text models (TBM (Ludan et al., 2023)) and image models (EAC (Sun et al., 2023)), respectively.

## 2 PRELIMINARIES

In this section, we introduce the background knowledge and notations used in this paper.

**Machine Learning Models.** We consider a machine learning model as a black-box function $f$ that maps an input vector $\boldsymbol{x}$ to an output scalar $f(\boldsymbol{x})$. Formally, we let $f : \mathbb{X} \to \mathbb{R}$, where $\mathbb{X}$ is the input domain. For models that take fixed-dimension inputs, let $\mathbb{X} = \mathbb{R}^n$. For models capable of handling inputs of arbitrary dimensions, let $\mathbb{X} = \cup_{i=1}^{\infty} \mathbb{R}^i$. Let $\boldsymbol{x}_i$ denote the $i$-th feature value of $\boldsymbol{x}$.

**Local Model-Agnostic Explanation Techniques.** A local model-agnostic explanation technique $t$ takes a model $f$ and an input $\boldsymbol{x}$, and generates a local explanation $g_{f,\boldsymbol{x}}$ to describe the behavior of $f$ around $\boldsymbol{x}$, i.e., $g_{f,\boldsymbol{x}} := t(f, \boldsymbol{x})$. $g_{f,\boldsymbol{x}}$ ($g$ for short) is an expression formed with **predicates**. Each predicate $p$ maps an input $\boldsymbol{x}$ to a binary value, i.e., $p : \mathbb{X} \to \{0, 1\}$, indicating whether $\boldsymbol{x}$ satisfies a specific condition.

As shown in Figure 4, existing local model-agnostic explanation techniques generate explanations following a similar workflow:

1. **Producing Predicates**: These techniques first generate **a set of predicates** $\mathbb{P}$ based on the input $\boldsymbol{x}$.

2. **Generating Samples**: The underlying **perturbation model** $t_{per}$ generates a set of samples $\mathbb{X}_s^b$ in **predicate representations**, where each sample $\boldsymbol{z}^b \in \mathbb{X}_s^b$ is a binary vector in $\{0, 1\}^d$ and $\boldsymbol{z}_i^b$ indicates whether the sample satisfies the $i$-th predicate in $\mathbb{P}$. The perturbation model then transforms the samples $\mathbb{X}_s^b$ back to the original input space to get $\mathbb{X}_s$ and $f(\mathbb{X}_s)$.

3. **Learning Explanation**: The underlying **learning algorithm** generates the local explanation $g_{f,\boldsymbol{x}}$ consisting of predicates in $P$ using $\mathbb{X}_s$ and $f(\mathbb{X}_s)$.

Mainstream local model-agnostic explanation techniques like Anchors, LIME, LORE, and Kernel SHAP, all follow this workflow. They use the same kinds of predicate sets and perturbation models, and use different learning algorithms to generate explanations with different properties. In the following, we introduce the main components of the explanation techniques.

**Predicate Sets.** Given an input $\boldsymbol{x}$, the corresponding predicate set $\mathbb{P}$ is defined as follows:

$$\mathbb{P} = \{p_i | i \in [1, d]\},$$

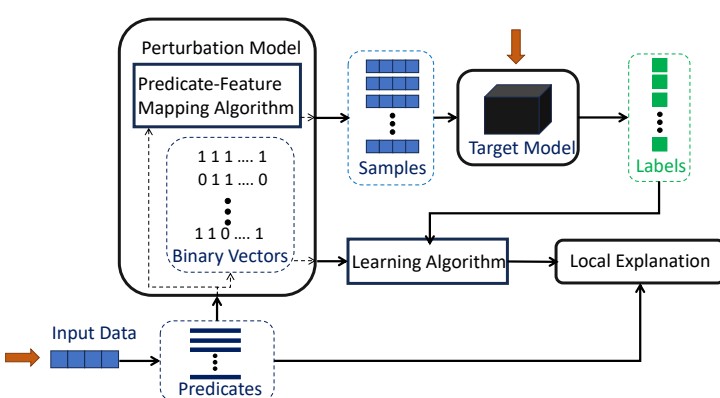

Figure 4: The workflow of generating explanations by a local model-agnostic explanation technique.

where $d$ is the number of predicates in $\mathbb{P}$, a hyperparameter set by users or according to the input data $\boldsymbol{x}$. Each $p_i$ is a **feature predicate** that constrains the value of a set of feature values in $\boldsymbol{x}$, i.e. $p_i(\boldsymbol{z}) : \bigwedge_{j \in \mathbb{A}_i} 1_{\mathrm{ran}(\boldsymbol{x}, j)}(z_j)$, where $\mathbb{A}_i$ is the set of indices of features that $p_i$ constrains. Specifically, $\{\mathbb{A}_1, \mathbb{A}_2, ..., \mathbb{A}_d\}$ forms a partition of $\{1, 2, \ldots, |x|\}$. Each $\mathrm{ran}(\boldsymbol{x}, j)$ is a set containing $x_j$, which is set according to the type of input data. For example, we can use $\mathrm{ran}(\boldsymbol{x}, j) = (x_j - \epsilon, x_j + \epsilon)$ for continuous values, and $\mathrm{ran}(\boldsymbol{x}, j) = \{x_j\}$ for categorical values. The predicate $p_i$ is a conjunction of indicator functions, each of which checks if a sample $z$ has a similar value to $x_j$ (i.e., $z_j \in \mathrm{ran}(\boldsymbol{x}, j)$).

**Predicate Representations.** The predicate representation $\boldsymbol{z}^b \in \{0, 1\}^d$ of a sample $\boldsymbol{z}$ is a binary vector where $\boldsymbol{z}_i^b = p_i(\boldsymbol{z})$.

**Perturbation Models.** The perturbation model $t_{per}$ first randomly selects $\mathbb{X}_s^b \subseteq \{0, 1\}^d$ as the predicate representations of the samples. Then, it transforms $\mathbb{X}_s^b$ back to the original input space to get $\mathbb{X}_s$. For each $\boldsymbol{z}^b \in \mathbb{X}_s^b$, a predicate-to-feature mapping function $t_{\mathrm{p2f}} : \{0, 1\}^d \to \mathbb{X}$ transforms $\boldsymbol{z}^b$ to $\boldsymbol{z}$ as follows: if $z_i^b = 1$, then for each $j \in \mathbb{A}_i$, set $z_j = x_j$; otherwise, set each $z_j$ to a masked value, or a random value sampled from $\mathrm{per}(\boldsymbol{x}, j) \backslash \mathrm{ran}(\boldsymbol{x}, j)$, where $\mathrm{per}(\boldsymbol{x}, j)$ is a perturbation range with $\mathrm{per}(\boldsymbol{x}, j) \supset \mathrm{ran}(\boldsymbol{x}, j)$.

**Learning Algorithms and Explanations.** The learning algorithm learns an understandable expression $g$ as an explanation. In Anchors, $g$ is a conjunction of predicates that provides a sufficient condition for producing $f(\boldsymbol{x})$ as output, i.e., $f(\boldsymbol{z}) = f(\boldsymbol{x})$ if $g(z) = 1$, and $g(z) = \bigwedge_{p \in \mathbb{Q}} p(z)$, where $\mathbb{Q}$ is selected by KL-LUCB algorithm (Kaufmann & Kalyanakrishnan, 2013). In LIME and Kernel SHAP, $g(z) = \sum_{i=1}^d w_i p_i(z) + w_0$, where $w_i$ is the weight of $p_i$, which is learned by their underlying regression algorithms. LORE first learns a decision tree with building systems like Yadt (Ruggieri, 2004), then extracts a sufficient condition to obtain $f(\boldsymbol{x})$ and some counterfactual rules from the tree as explanations. The sufficient condition is similar to Anchors, while each of the counterfactual rules is in the form of $f(\boldsymbol{z}) = y$ if $g(z) = 1$, where $y \neq f(\boldsymbol{x})$ and $g(z) = \bigwedge_{p \in \mathbb{Q}} p(z) \wedge \bigwedge_{p \in \mathbb{C}} \neg p(z)$, and $\mathbb{Q}$ and $\mathbb{C}$ are extracted from the decision tree.

**An Example.** For a text input *I love this movie so much*, these techniques can let each $p_i$ constrains only one feature, and produce six predicates in the form of $p_i(z) := \mathbf{1}_{z_i = x_i}$. For another text input $z = $ *I love this [MASK] so [MASK]*, the predicate representation of $z$ is $p_1(z)\, p_2(z)\, p_3(z)\, p_4(z)\, p_5(z)\, p_6(z) = 1\,1\,1\,0\,1\,0$. The perturbation model will generate samples in predicate representation, then $t_{p2f}$ will transform samples back to the origin input space. For example, a sample $0\,1\,0\,1\,1\,1$ is generated and $t_{p2f}$ maps it to *[MASK] love [MASK] movie so much*. Consequently, these techniques will use the output of these samples and the samples' predicate presentation to learn an expression, and build the explanation with the predicates.

Limited by the predicate sets and perturbation models, existing local model-agnostic explanation techniques can only generate explanations based on the constraints of feature values, which limits their effectiveness in explaining the behavior of the model.

# 3 THE CONLUX FRAMEWORK

In this section, we propose ConLUX, a general framework to provide concept-based local explanations based on existing local model-agnostic explanation techniques without significantly changing their core components.

We introduce ConLUX in three steps: 1) defining concept-based local explanations and concept predicates, 2) showing the modifications to the explanation techniques, and 3) demonstrating the augmented workflow.

## 3.1 CONCEPT-BASED LOCAL EXPLANATIONS

As we discussed in Section 2, though the form of the explanations varies, they are all built from predicates in $\mathbb{P}$. Elevating the predicates to concept level is the key to providing concept-based explanations.

Definition of high-level concepts varies, as Molnar (2020) mentions, "A concept can be any abstraction, such as a color, an object, or even an idea." Here, to provide explanations that are easier to understand by end-users, we define a concept predicate as follows:

**Definition 1** (Concept Predicate). *Given an input $x$, a concept predicate $p^c$ is a function that maps $x$ to a binary value, i.e., $p^c : \mathbb{X} \to \{0, 1\}$, and satisfies the following properties:*

1. ***Descriptive****: The concept predicate $p^c$ can be concisely and intuitively described in natural language.*

2. ***Human Evaluable****: The truth of $p^c(x)$ can be readily assessed by a human user.*

The preceding two properties ensure that the concepts are easy to understand. Here, we provide two examples of concept predicates for text and image models in the following:

**Examples.** For text models, we can define a concept predicate as follows:

- Concept Name: Poor Visual Effects and Cinematography

- Description: The input text mentioned that the visual effects and cinematography are lacking, failing to create an appealing aesthetic.

For image models, we use objects in an image to define a concept predicate. As shown in Figure 2, we can easily describe the concept predicate as "there is a punching bag in the image", "there is a kid in the image", etc.

We then define a concept-based local explanation as follows:

**Definition 2** (Concept-Based Local Explanation). *A concept-based local explanation $g^c_{f,x}$ is an expression formed with concept predicates to describe the behavior of $f$ around $x$.*

As existing local explanation model-agnostic techniques provide various kinds of explanations like attributions, sufficient conditions, and counterfactuals, ConLUX can elevate all these explanations to concept level and provide users a unified interface to obtain various kinds of explanations with a single click. Additionally, We denote such a set of various kinds of explanations as a **ConLUX unified explanation**, which provides higher fidelity and offers a more comprehensive view than a single form of explanation.

## 3.2 AUGMENTING EXPLANATION TECHNIQUES

As shown in Figure 4, to produce concept predicates, we should first extract high-level concepts based on the input $x$; to provide explanations at concept level, we should replace the feature predicates in $\mathbb{P}$ with concept predicates; to capture the local behavior of the target model at concept level, we should extend the perturbation model to generate samples by changing high-level concepts.

**Producing Concept Predicates.** We use large pre-trained models to provide high-level concepts based on the input $x$ and the target task. For text models, following the approach of Ludan et al. (2023), we provide GPT-3.5 with task-specific information, the given input, the corresponding output, and several in-context learning examples to generate candidate concepts. These concepts are then evaluated on the input $x$ to construct the concept predicate set.

For image models, we refer to Sun et al. (2023) to use SAM to detect objects in the image.

Consequently, ConLUX defines concept predicates (denoted as $p^c$) using the extracted concepts, and replaces the feature predicates set $\mathbb{P}$ with the concept predicates set $\mathbb{P}^c$.

**Performing Concept-Level Perturbation** The extended perturbation model $t_{per}^c$ generates samples in concept-level representation and $t_{p2f}^c : \{0,1\}^{|\mathbb{P}^c|} \to \mathbb{X}$ transforms the samples back to the original input space. Different from the $t_{per}$ simply decides whether to mask a feature value, $t_{per}^c$ changes high-level concepts at feature level, which is more complex. Therefore, $t_{per}^c$ is usually a more sophisticated model. Here, for text models, we use Llama 3.1 to perform the concept-feature mapping; for image models, since each object is still a set of pixels, we can use the same transformation as $t_{per}$. As the effectiveness and faithfulness of the text perturbation are not straightforward, we conduct an experiment to demonstrate this, as detailed in Appendix C.

### 3.3 ConLUX-Augmented Workflow

The ConLUX augments the workflow in Figure 4 as follows: it first extracts high-level concepts based on the input $x$ and the target task, then follows a similar workflow as their vanilla versions, but replaces the predicate set $\mathbb{P}$ with $\mathbb{P}^c$ and the perturbation model $t_{per}$ with $t_{per}^c$. Therefore, the ConLUX-augmented techniques can capture the local behavior of the target model at the concept level, and provide concept-based local explanations.

More details about the implementation of ConLUX can be found in Appendix A.

## 4 Empirical Evaluation

In this section, we demonstrate the generality of ConLUX, its improvement of explanation fidelity, and the fidelity of ConLUX unified explanations by empirical evaluation. We show the generality of ConLUX by applying it to four mainstream local model-agnostic explanation techniques: Anchors, LIME, LORE, and Kernel SHAP (KSHAP for short), which provide three types of explanations: sufficient conditions, counterfactuals, and attributions. We apply them to explain various text and image models. We show the improvement of explanation fidelity by comparing the vanilla feature-level explanations with ConLUX-augmented the concept-based explanations. Moreover, we compare the fidelity of ConLUX unified explanations with two state-of-the-art concept-based explanation techniques: Textual Bottleneck Model (TBM) (Ludan et al., 2023) for text models and Explain Any Concept (EAC) (Sun et al., 2023) for image models.

### 4.1 Experimental Setup

We chose sentiment analysis as the target task for text models, and image classification as the target task for image models.

**Sentiment Analysis.** Sentiment analysis models take a text sequence as input and predict if the text is positive or negative, i.e. $f : \mathbb{X} \to \{0,1\}$, where $\mathbb{X} := \bigcup_{i=1}^{\infty} \mathbb{W}^i$ is the input domain, and $\mathbb{W}$ is the vocabulary set. We used a pre-trained BERT (Morris et al., 2020) and Llama3.1 to predict the sentiment of 200 movie reviews from the Large Movie Review Dataset (Maas et al., 2011), and explained the local behavior of the models around each input text. For vanilla techniques, we followed the settings described in Section 2. For ConLUX-augmented techniques, we set the number of concept predicates to 10, used GPT-3.5 (Brown et al., 2020) to extract high-level concepts, and Llama3.1 to perform the predicate-to-feature mapping. For TBM, we applied it to explain the same 200 movie reviews with its default settings.

**Image Classification.** Image classification models take an image as input and predict the category of the image, i.e. $f : \mathbb{X} \to \{0, 1, ..., m\}$, where $m$ is the number of categories, $\mathbb{X} := \mathbb{R}^{3 \times h \times w}$ is the input domain, with $h$ and $w$ being the height and width of the image. We used a pre-trained YOLOv8, Vision Transformer (ViT) (Oquab et al., 2023; Darcet et al., 2023), and ResNet-50 (He et al., 2016) to predict the category of 1000 images from the ImageNet dataset (Deng et al., 2009), and explained the local behavior of the models around each input image. For vanilla techniques, we followed LIME to use Quickshift algorithm (Jiang et al., 2018) to obtain the superpixels, and used these super-pixels as predicates. For ConLUX-augmented techniques, we used SAM (Kirillov et al., 2023) to detect objects in the images, and used these objects as predicates. For EAC, we applied it to explain the same 1000 images with its default settings.

To evaluate the fidelity of ConLUX unified explanations, as the combination of multiple forms of explanation provides more fidelity than a single form, we used the combination ConLUX-augmented KSHAP and LORE explanations as local surrogate models. Specifically, if an input is covered by LORE's rule, we use the LORE output; otherwise, we use the KSHAP explanation.

More details can be referred to Appendix B.

## 4.2 FIDELITY EVALUATION

### 4.2.1 EVALUATION METRICS

Fidelity reflects how faithfully the explanations describe the target model. As these techniques provide explanations in different forms, we used different metrics to evaluate their fidelity.

Following the setup in the original papers of Anchors and LORE, we used **coverage** and **precision** as fidelity metrics (which are named differently in the LORE paper). Given a target model $f$, an input $\boldsymbol{x}$, and a distribution $D_{\boldsymbol{x}}$ derived from the perturbation model, and the corresponding explanation $g$, we defined the coverage as $\text{cov}(\boldsymbol{x}; f, g) = \mathbb{E}_{\boldsymbol{z} \sim D_{\boldsymbol{x}}}[g(\boldsymbol{z})]$, which indicates the proportion of inputs in the distribution that match the rule; we defined the precision as $\text{prec}(\boldsymbol{x}; f, g) = \mathbb{E}_{\boldsymbol{z} \sim D_{\boldsymbol{x}}}[\mathbf{1}_{f(\boldsymbol{z})=y}|g(\boldsymbol{z})]$, where y is the consequence of the rules in $g$ with $y = f(\boldsymbol{x})$ for factual rules and $y \neq f(\boldsymbol{x})$ for counterfactual rules. Precision indicates the proportion of covered inputs that $g$ correctly predicts the model outputs.

As LIME and KSHAP are attribution-based local surrogate, we used *Area Over most relevant first perturbation curve* (AOPC) (Samek et al., 2016; Modarressi et al., 2023), and $\text{accuracy}_{\text{a}}$ as fidelity metrics (Balagopalan et al., 2022; Yeh et al., 2019a; Ismail et al., 2021). Given a target model $f$, an input $\boldsymbol{x}$, its corresponding model output $y = f(\boldsymbol{x})$, their corresponding explanation $g$, and $x^{(k)}$ that is generated by masking the $k\%$ most important predicates in $\boldsymbol{x}$, AOPC and $\text{accuracy}_{\text{a}}$ are defined as follows:

- **AOPC:** Let $\text{AOPC}_k = \frac{1}{|\mathbb{T}|} \sum_{\boldsymbol{x}}^{\mathbb{T}} p_f(y|\boldsymbol{x}) - p_f(y|\boldsymbol{x}^{(k)})$, where $p_f(y|\boldsymbol{x})$ is the probability of $f$ to output $y$ given the input $\boldsymbol{x}$, and $\mathbb{T}$ is the set of all test inputs. $AOPC_k$ indicates the average change of the model output when masking the $k\%$ most important predicates. A higher $AOPC_k$ indicates a better explanation. We calculate the AOPC curve by varying $k$ from 0 to 100.
- **Accuracy$_{\text{a}}$:** Accuracy$_{\text{a}}$ indicates the proportion of inputs among all $\boldsymbol{x}^{(k)}$ that the target model gives the same output as the original input $\boldsymbol{x}$, i.e. $\mathbb{E}(f(\boldsymbol{x}^{(k)}) = f(\boldsymbol{x}))$. Specifically, $\text{accuracy}_{\text{a}}$ is different from the standard accuracy, and a lower $\text{accuracy}_{\text{a}}$ indicates a better explanation.

Specifically, we only considered the predicates that positively contribute to $f(\boldsymbol{x})$, and we did not use AOPC when explaining Llama 3.1, as it does not directly provide the probability for each output token.

For TBM, EAC, and ConLUX unified explanations, considering that they can all serve as local surrogate models, i.e. $g : \mathbb{X} \to \mathbb{R}$, we defined the metrics as follows: Given a target model $f$, an input $\boldsymbol{x}$, a perturbation distribution $\mathbb{D}_{\boldsymbol{x}}$, and their corresponding explanation $g$, a performance metric $L$ (e.g. accuracy, F1 score, MSE, etc.), we define the (in-)fidelity as $E_{\boldsymbol{z} \sim \mathbb{D}_{\boldsymbol{x}}} L(f(\boldsymbol{z}), g(\boldsymbol{z}))$, which indicates the performance of using $g$ to approximate $f$. Here, we used the accuracy as the performance metric. Specifically, considering the complexity of the original task, we reduced the image classification task for local surrogates to predicting whether the target model $f$ assigns the same classification to $\boldsymbol{x}'$ as it does to $\boldsymbol{x}$.

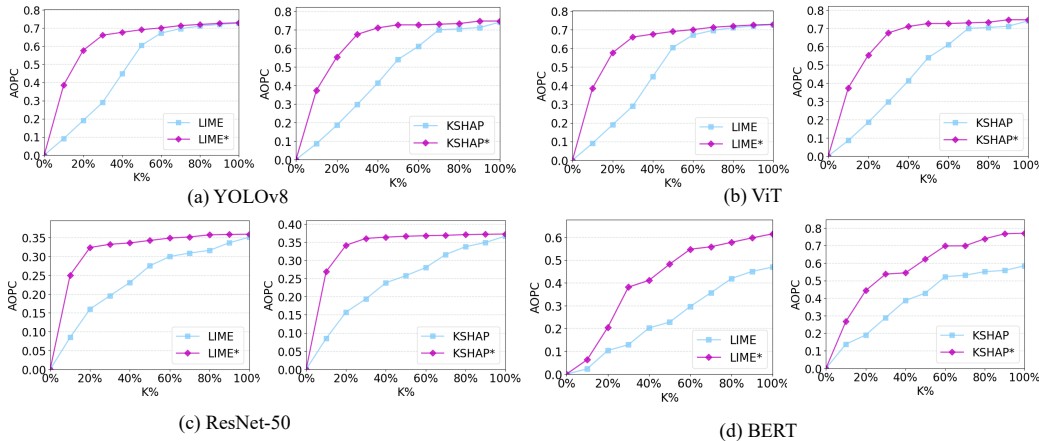

Figure 5: AOPC upon masking the K% most important predicates. We use LIME, Kernel SHAP, and their ConLUX-augmented version to explain YOLOv8, ViT, and Resnet-50 on the image classification task and BERT on the sentiment analysis task.

Table 1: Average coverage and precision (higher are better) of Anchors, LORE, and their ConLUX-augmented versions (denoted as Anchors* and LORE*) on two sentiment analysis models and three image classification models.

| Models | Coverage (%) ↑ | | | | Precision (%) ↑ | | | |
|---|---|---|---|---|---|---|---|---|
| | Anchors | Anchors* | LORE | LORE* | Anchors | Anchors* | LORE | LORE* |
| Llama 3.1 | 4.9 | **22.5** | 2.3 | **21.3** | 81.2 | **94.2** | 64.3 | **76.9** |
| BERT | 5.3 | **24.4** | 3.2 | **20.3** | 78.2 | **91.0** | 65.3 | **79.4** |
| YOLOv8 | 28.6 | **30.9** | 20.8 | **24.8** | 84.3 | **98.2** | 87.8 | **92.2** |
| ViT | 24.6 | **28.2** | 21.3 | **23.8** | 88.7 | **98.2** | 89.6 | **95.6** |
| ResNet-50 | 28.0 | **30.5** | 20.1 | **29.7** | 89.3 | **99.4** | 85.8 | **92.6** |

### 4.2.2 EVALUATION RESULTS

Table 1 shows the fidelity of Anchors, LORE, and their ConLUX-augmented versions. ConLUX improves the average coverage of Anchors and LORE by 9.0% and 10.4%, and the average precision by 11.9% and 8.7%, respectively. Figure 5 and Table 2 show the fidelity of LIME, KSHAP, and their ConLUX-augmented versions. Figure 5 shows the AOPC curve of LIME and KSHAP. Each AOPC curve of ConLUX-augmented versions is higher than the vanilla counterpart. Table 2 shows the average AOPC and $\text{accuracy}_a$. ConLUX improves the average AOPC by 0.122 and 0.145, and decreases the average $\text{accuracy}_a$ by 21.6% and 22.8%, for LIME and KSHAP, respectively. We do paired t-tests for each setup that only differs on whether to apply ConLUX, to show the statistical significance of the improvement. The p-value is all less than 0.01, which indicates with over 99% confidence the improvement is significant.

We also compared ConLUX unified explanations with two state-of-the-art concept-based task-specific explanation techniques: TBM for text tasks and EAC for image tasks. Table 3 shows the fidelity of TBM, EAC, and ConLUX unified explanations. ConLUX helps two classic local model-agnostic techniques to achieve 5.75% and 4.9% more accuracy than TBM and EAC.

## 5 RELATED WORK

Our work is related to model-agnostic explanation techniques and concept-based explanation techniques.

Model-agnostic explanation techniques consider target models as black boxes and provide explanations without requiring any knowledge of the model's internal structure. Existing Model-agnostic explanation techniques provide different types of explanations, such as feature importance (Lund-

Table 2: Average AOPC and $\text{accuracy}_a$ (higher AOPC and lower $\text{accuracy}_a$ are better) of LIME, KSHAP, and their ConLUX-augmented versions (denoted as LIME* and KSHAP*) on two sentiment analysis models and three image classification models.

| Models | AOPC ↑ | | | | $\text{Accuracy}_a$ (%) ↓ | | | |
|---|---|---|---|---|---|---|---|---|
| | LIME | LIME* | KSAHP | KSAHP* | LIME | LIME* | KSAHP | KSAHP* |
| Llama 3.1 | – | – | – | – | 82.3 | **49.8** | 72.1 | **45.4** |
| BERT | 0.243 | **0.456** | 0.379 | **0.553** | 75.7 | **47.7** | 60.3 | **40.2** |
| YOLOv8 | 0.401 | **0.474** | 0.433 | **0.590** | 14.8 | **5.0** | 33.1 | **6.9** |
| ViT | 0.469 | **0.598** | 0.454 | **0.611** | 21.1 | **4.9** | 25.7 | **9.0** |
| ResNet-50 | 0.232 | **0.306** | 0.233 | **0.323** | 36.3 | **14.6** | 33.1 | **8.6** |

Table 3: Average accuracy (higher accuracy is better) of TBM, EAC, and ConLUX unified explanations on two sentiment analysis models and three image classification models.

| Methods | Accuracy (%) ↑ | | | | |
|---|---|---|---|---|---|
| | Llama 3.1 | BERT | YOLOv8 | ViT | ResNet-50 |
| TBM | 89.6 | 81.4 | – | – | – |
| EAC | – | – | 56.6 | 53.4 | 57.7 |
| ConLUX | **94.7** | **87.8** | **61.3** | **59.6** | **61.5** |

berg & Lee, 2017; Ribeiro et al., 2016; Tan et al., 2023; Shankaranarayana & Runje, 2019), decision rules (Ribeiro et al., 2018; Guidotti et al., 2018; Dhurandhar et al., 2018), counterfactuals (Wachter et al., 2018; Guidotti et al., 2018), and visualizations (Goldstein et al., 2015; Friedman, 2001; Apley & Zhu, 2020). However, to our knowledge, all existing model-agnostic explanation techniques provide explanations at feature levels (Zhang et al., 2021c). Basic feature-based explanations are usually worse in aligning with either the decision-making process of the model or end-users (Ghorbani et al., 2019a; Sun et al., 2023; Kim et al., 2018), which makes these explanations unfaithful and hard to understand.

Concept-based explanation techniques provide explanations in terms of high-level concepts, which align with the decision-making process of the model better and are more interpretable to end-users. To our knowledge, existing concept-based explanation techniques are all model-specific or task-specific (Poeta et al., 2023b). We categorize them into three groups: (1) techniques that extract concepts from the model's internal structure (Zhang et al., 2021b; Yeh et al., 2020; 2019b; Cunningham et al., 2023; Ghorbani et al., 2019b; Crabbé & van der Schaar, 2022; Fel et al., 2023), which are limited to specific types of models, (2) techniques that use external knowledge to define concepts (El Shawi, 2024; Widmer et al., 2022), which are limited to specific types of tasks since their methods based on the knowledge for a specific task, and (3) techniques that use pre-trained models to extract concepts (Ludan et al., 2023; Sun et al., 2023). Ludan et al. (2023) propose TBM, which is a surrogate model specifically designed for text data, while Sun et al. (2023) propose EAC, which also utilizes internal information of the target model. Therefore, these techniques are only for specific types of tasks. In addition, these techniques mainly focus only on attributions which limits their use cases (Poeta et al., 2023b).

## 6 CONCLUSION

We have proposed ConLUX, a general framework that automatically extracts high-level concepts and incorporates them into existing local model-agnostic explanation techniques to provide concept-based explanations, which are more faithful and easier to understand by end-users. ConLUX offers unified explanations that combine attributions, sufficient conditions, and counterfactuals. This satisfies diverse user needs and fills the current gap in concept-based explanations, which lack forms beyond attributions. ConLUX achieves this by utilizing large pre-trained models to extract high-level concepts, elevating language components from feature level to concept level, and extending perturbation models to sample in the concept space. We have instantiated ConLUX on Anchors, LIME, LORE, and Kernel SHAP, and provide unified explanations. We have constructed empirical evaluations to demonstrate the effectiveness of ConLUX.

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

# A  THE CONLUX FRAMEWORK (CONTINUED)

In this section, we introduce the details of incorporating ConLUX into existing local explanation methods.

We first follow Section 3 to introduce how we extend each part for text models in detail.

## A.1  PRODUCING CONCEPT

ConLUX provides predicates that describe high-level concepts by utilizing a large pre-trained model. Here, we describe the step to extract concept from text and image data in detail.

**Text Data.** We use GPT-3.5 to produce concept-level predicates in two step. First, we let $GPT-3.5$ generate the concepts that are important to the current task with a prompt as follows:

> Now you are an expert at writing movie reviews, please tell me from which per-spectives can you evaluate a movie.

Then we let the $GPT-3.5$ to refine the predicates based on the current input and its similar sentences, and format the concepts. Here, we referred to the format defined in TBM (Ludan et al., 2023). The prompt is as follows:

> Here we are presented with a text dataset accompanied by labels, and our objective is to identify a concept in the text that correlates with these labels. The task is to ....., we have known the following concepts are important in this task. [concepts]
>
> In additionally, you should refine the concept to make sure that concepts can be used to correctly classify the following examples: [texts labels]
>
> Then you are given examples of concepts across various datasets. please give me the concepts following their format:
>
> Example 1:
>
> "Concept Name": "explicit language",
>
> "Concept Description": "'Explicit language' refers to the use of words, phrases, or expressions that are offensive, vulgar, or inappropriate for general audiences. This may include profanity, obscenities, slurs, sexually explicit or lewd language, and derogatory or discriminatory terms targeted at specific groups or individuals.",
>
> "Concept Question": "What is the nature of the language used in the text?",
>
> "Possible Responses": ["explicit", "strong", "non-explicit", "uncertain"],
>
> "Response Guide":
>
> "explicit": "The text contains explicit language, such as profanity, obscenities, slurs, sexually explicit or lewd language, or derogatory terms targeted at specific groups or individuals.",
>
> "Strong": "The text contains strong language but not explicit language, it may contain terms that some viewers might find mature.",
>
> "non-explicit": "The text is free from explicit language and is appropriate for general audiences.",
>
> "uncertain": "It is difficult to determine the nature of the language used in the text or if any explicit terms are used." ,
>
> "Response Mapping":
>
> "explicit": 2,
>
> "strong":1,
>
> "non-explicit": -1,
>
> "uncertain": 0
>
> Example 2:
>
> ......
>
> Now, please give me your formatted concepts:

Then we use the Response Guides to produce local concepts.

## A.2 CONCEPT-FEATURE MAPPING

We the the following prompt for predicate-feature mapping:

> Generate a sentence similar to a given sentence from the domain of dataset. The dataset's description is that .
>
> The generated sentence satisfies given concepts. Before generating the sentence, carefully read the description of each concept to understand the properties that the generated sentence must satisfy, think about how the sentence satisfies the concepts first, and then create the sentence that satisfies the concepts.
>
> Format your response as a JSON with string keys and string values. Below is an example of a valid JSON response. The JSON contains keys thoughts, and the answer. End your response with ###
>
> — Concepts: 1. Concept 1 2. Concept 2 ...
>
> Response JSON: "thoughts": "In this section, you explain which snippets in your text support the concepts. Be as objective as possible and ignore irrelevant information. Focus only on the snippets and avoid making guesses.", "answer": "A sentence that satisfies the concepts." ###
>
> Two examples of this task being performed can be seen below. Note that the answer should be in 5 to 20 words and should be a single sentence.
>
> Example 1:
>
> Concepts: 1. The plot of the text is exciting, captivating, or engrossing. It may have unexpected twists, compelling conflicts, or keep the reader eagerly turning pages. 2. The characters in the movie are portrayed in a realistic and convincing manner. Their actions, dialogue, emotions, motivations, and development feel authentic and relatable, making them believable to the audience. 3. The narrative structure of the text is confusing or unclear, making it difficult to follow or comprehend the events happening within the story. 4. The text introduces some original elements or takes minor risks in the plot development, but overall, it lacks a truly unique or innovative narrative.
>
> Response JSON: "thoughts": "The snippet 'the silly and crude storyline' mentions a storyline that is described as silly and crude mentions a 'silly and crude storyline' which indicates a lack of creativity and reliance on clichéd plot devices, satisfying the concept of some originality and inventive plot development. The snippet 'the real issues tucked between the silly and crude storyline', mentions a contrast between real issues and a silly and crude storyline, indicating a potentially confusing narrative structure, satisfying the concept of a confusing narrative structure. This also mentions that it has real conflict inside, which satisfies the concept of an exciting plot. This snippet also mentions mentions 'real issues' which indicates that the characters are portrayed in a realistic and convincing manner, satisfying the concept of realistic and convincing characters." "answer": "it's about issues most adults have to face in marriage and i think that's what i liked about it – the real issues tucked between the silly and crude storyline." ###
>
> Example 2: .... ###
>
> Perform the task below, keeping in mind to limit the response to 5 to 20 words and a single sentence. Return a valid JSON response ending with ###
>
> Concepts:
>
> Response JSON:

## B EXPERIMENT SETTINGS (CONTINUED)

We experimented on two machines, one with an Intel i9-13900K CPU, 128 GiB RAM, and RTX 4090 GPU, and another with Intel(R) Xeon(R) Silver 4314 CPU, 256GiB RAM, and 4 RTX 4090 GPUs.

To measure the fidelity improvement brought by ConLUX, we keep all hyperparameters the same for both vanilla and augmented methods.

For LIME and KSHAP, we set the number of sampled inputs to 1000 except for explaining Llama 3.1.

For Anchors, we follow the default settings.

For LORE, we set $ngen = 5$.

For the LLama3.1 model, when applying it to the sentiment analysis task, we simply use the following prompt:

> From now on, you should act as a sentiment analysis neural network. You should classify the sentiment of a sentence into positive or negative. If the sentence is positive, you should reply 1. Otherwise, if it's negative, you should reply 0. There may be some words that are masked in the sentence, which are represented by <UNK>. The input sentence may be empty, which is represented by <EMPTY>. You will be given the sentences to be classified, and you should reply with the sentiment of the sentence by 1 or 0.
> There are two examples:
> Sentence:
> I am good
> Sentiment:
> 1
> Sentence:
> The movie is bad.
> Sentiment:
> 0
> You must follow this format. Then I'll give you the sentence. Remember Your reply should be only 1 or 0. Do not contain any other content in your response. The input sentence may be empty.
> Sentence:
> {The given sentence}
> Sentiment:

## C  TEXT PERTURBATION FAITHFULNESS EXPERIMENT

As demonstrated by Ludan et al. (2023), large language models (LLMs) can verify whether an instance satisfies a given concept. Building on this, we conduct an experiment to evaluate the consistency of LLM-based perturbations. Specifically, we use LLMs to assess whether the applied perturbations successfully alter the intended concept. For each sentence, we generate 100 random perturbations and verify if the concepts in the generated sentences align with the expected changes. Our results indicate that Llama3.1, the large model employed in our fidelity experiments, achieves concept-level perturbation accuracy exceeding 99

