# OpenReview forum: "ConLUX: Concept-Based Local Unified Explanations"
_ICLR.cc/2025/Conference — Submitted to ICLR 2025_

### Official Review · Reviewer_B3KS · 2024-10-27

**Soundness:** 3
**Presentation:** 3
**Contribution:** 2
**Rating:** 6
**Confidence:** 4

**Summary:**

The paper proposes a concept based local explanation method ConLUX that is model agnostic. The authors essentially propose a modality specific concept representations of inputs (concept predicates). These representations also readily provide a procedure to perform perturbation on the predicates and subsequently the input. Combining these two, the method is able to augment the traditional model-agnostic approaches to provide concept based explanations. The authors provide experiments on text (sentiment prediction) and images (classification) with multiple black-box models and explanation techniques and essentially show a clear improvement in terms of various forms of fidelity.

**Strengths:**

1. The problem setup, what the authors want to solve, and why, is quite clear.
2. The core idea of using a concept-friendly representation for input and combining it with black-box explanation methods is simple and its positive implications are easy to see.
3. The experiments are reasonably strong. They cover both text, images, and on multiple black-box models, with positive results in all cases. Also, via the existing black-box explainers, the method can generate different types of explanations (attribution, counterfactual etc.)
4. While it has its own weaknesses, the proposal to build visual concept predicates is a principled way that could be readily validated by a user if the concept extraction is incorrect. This aspect of simplicity should positively reflect in its application

**Weaknesses:**

1. Twice the authors describe (Fel et al. 2023) as using external knowledge to learn concepts. To my understanding, this is a wrong description. They propose a unified class of methods based on dictionary learning that is completely unsupervised.
2. Typos/Errors:
    * Rednet (line 448)
    * line 351 should not be in past tense
    * Table 3 caption does not correspond to the table content
3. The method seems only capable to extract coarse visual concepts. Also it can only admit concepts that can be represented as a segmentation mask. In case of text concept predicates, potential risk of some issues arising from using language models for concept detection and predicate-feature mapping.
4. I felt a lack of examples/illustrations of visual explanations and any deeper qualitative insights the authors might have.

**Questions:**

1. I am not completely convinced by the strategy of using language models for text concept predicates. While I assume it probably provides the best performance and our generally more than good enough for text-only tasks, they could still be prone to hallucinations in terms of incorrect concept extraction or during generations for perturbation. Did you consider some other method (maybe traditional topic modelling approaches) to validate its concept detection or perturbation outputs. In this sense I like the visual predicates lot more than textual ones.

2. I wonder if you considered an ACE (Ghorbani et al.) + LIME method to compare ConLUX against a concept based reference where you use activation space of an external encoder to cluster superpixels instead of the original model. The concepts can be defined as clusters of superpixels, as in original method.

Overall, comparing the strengths and weaknesses, I find the method to be just sound and strong enough that I would tilt slightly towards acceptance.

---

> ### Author Response · Authors · 2024-11-20
> **Rebuttal by Authors**
>
> Thanks for your review!
>
> Our responses to your concerns are as follows:
>
> **Typos and Errors:**
> We have uploaded our edited paper.
>
> **Concept for image data:**
> As you said, in our experiments, we use concepts represented as segmentation masks. However, we would like to clarify that this is just one instantiation of ConLUX-generated explanations. ConLUX is a framework that accepts different kinds of concepts, as long as the pre-trained models can obtain concepts and perform the bidirectional mapping between concept level and feature level. For example, it is possible to use more abstract concepts represented by nature language with models like BLIP-2(Li et al., 2023), and perform the perturbations by models like stable diffusion(Rombach et al., 2021).
>
> **Concept extraction and perturbation for text data:**
> In our text experiments, we employ a workflow similar to TBM (Ludan et al., 2023) to generate concepts. In the TBM paper, the quality of concepts and the ability of large language models (LLMs) to map feature-level data to concept-level representations have been validated through human evaluation.
> For perturbation, we propose to perform concept-level perturbation by an LLM, and we admit it's worth verifying the consistency of concept-level perturbation. As LLMs have been proved to be able to check whether an instance satisfies a given concept, we are conducting an experiment to evaluate the consistency of LLM-based perturbations. Specifically, we use LLMs to verify whether the perturbation alters the concept as intended. We will report these results in a subsequent update.
>
> **ACE + LIME as a baseline:**
> We do not believe ACE + LIME should be considered a baseline for our framework.
> - First, ConLUX is designed as a local explanation technique that takes a target model and an input to generate a local explanation. In contrast, ACE is claimed to be a global explanation method that explains an entire class and requires a set of images from that class to extract concepts.
> - Second, ACE segments images at multiple resolutions, meaning a concept does not correspond to a specific area of the image but to multiple segments at different resolutions. If we simply use the union of these segments to represent a concept, this can lead to overlapping areas between concepts, making it difficult to perturb a specific concept by simply masking the corresponding area.
> - Lastly, extending existing local methods to concept-level explanations is a key contribution of our paper. If the issues of concept discovery and perturbation can be addressed without pre-trained large models, then the augmented LIME can be considered a partial instantiation of ConLUX.

---

> ### Comment · Reviewer_B3KS · 2024-11-23
> **Response to Author Rebuttal**
>
> Thank you for the response.
>
> I do not fully agree with the first two points about ACE+LIME. The use of ACE for this baseline would be mainly for extract concepts. It should be completely reasonable for your method to provide local explanations but assume an initial set of images to source you with concepts. About perturbation too, either you can consider a simpler version with superpixel extraction at a single resolution or while perturbing with concepts you can take union of superpixels associated with any perturbed concept and not in any present concept.
>
> However, because I agree with your last point I think these concerns of mine are irrelevant. It should be fine if any such system was seen as an instantiation for your method. So it is ok for me if you didn't consider such a system as a baseline but part of your own method.
>
> The idea for using LLM/MLLMs and diffusion models is also interesting, although I don't think there were any experiments showing it.
> While this is not a strong enough concern for me, in general the image explanation experiments would strengthen if you also considered more sophisticated strategies for concept extraction, any that could extract concepts at a finer scale.

---

> ### Author Response · Authors · 2024-12-03
>
> Thanks again for your feedback.
>
> We have conducted the experiment to validate the faithfulness of LLM perturbation in our text experiments. Specifically, we test whether Llama-3.1, the LLM used in our experiments, can generate sentences with the expected concepts. The results show that Llama-3.1 generates sentences as expected with over 99% accuracy.

---

### Official Review · Reviewer_CduZ · 2024-11-03

**Soundness:** 3
**Presentation:** 2
**Contribution:** 2
**Rating:** 3
**Confidence:** 4

**Summary:**

- The paper introduces ConLUX a framework designed to enhance model-agnostic explanation methods by transforming traditional feature-level explanations into concept-level ones.
- The authors argue that mainstream model-agnostic explanation techniques often provide explanations based on low-level features that don't align well with model decision processes or user understanding so ConLUX elevates explanations to the concept level.
- The framework applies ConLUX to different explanation methods (LIME, Anchors, LORE, and Kernel SHAP) across text and image models.

**Strengths:**

- ConLUX shows promise in being adaptable to multiple existing explanation methods (LIME, SHAP, Anchors, LORE), broadening its application across varied model types.
- ConLUX aims to make model behaviors more intuitive and accessible for end-users, addressing a limitation in feature-based explanations.

**Weaknesses:**

**Major**
- One key weakness of ConLUX I felt was that since it shifts to a concept-level perturbations, which are broader and may disrupt the local fidelity of explanations. Unlike small feature-level adjustments (e.g., word or pixel changes in LIME), concept-level changes can alter the input more drastically, potentially leading to explanations that do not accurately reflect the model’s behavior around the specific input instance. To investigate whether these concept-level perturbations maintain local fidelity, I suggest running controlled experiments comparing concept-level and feature-level perturbations. For example, the authors could measure fidelity loss or gain across a gradient of perturbation scales, allowing for a comparison between the fidelity of feature-level and concept-level explanations.
- The paper relies on pre-trained models to extract high-level concepts but does not fully explore whether these concepts are consistently relevant across diverse domains. Variability in concept quality could impact the explanation's reliability. I recommend testing concept quality across datasets from different domains and introducing a metric or using existing ones like TCAV[1] to measure concept relevance and coherence within each domain.
- Since ConLUX relies on pre-trained models to extract high-level concepts, it inherits any biases present in these models. This reliance could skew the explanations based on the biases embedded in the pre-trained models, which might limit the fairness and reliability of the generated explanations.
- Observed improvement in fidelity metrics, such as AOPC, coverage, and precision, may partly result from the broader concept-level perturbations rather than genuinely enhanced explanation quality. Since larger perturbations at the concept level likely introduce more drastic changes to the model output, they could artificially inflate these scores, making the explanations appear more effective than they might be with finer, feature-level adjustments. To address this, the paper could benefit from controlled experiments using varying perturbation scales, comparing small and large concept-level shifts, to ensure that the fidelity improvements genuinely reflect enhanced interpretability rather than the impact of larger perturbations. I suggest implementing a normalized fidelity metric that adjusts for the magnitude of perturbations.

**Minor**
- A few typos in page 2 "ConLUX-agumented LIME" should be "ConLUX-augmented LIME"

**References**
1. Kim, Been, et al. "Interpretability beyond feature attribution: Quantitative testing with concept activation vectors (tcav)." International conference on machine learning. PMLR, 2018.

**Questions:**

- Don’t have much in terms of questions on the methodology itself, but a few conceptual issues stood out, as mentioned in the weaknesses.
- It’d be interesting to see if the authors could run additional experiments with different scales of perturbation to make sure these fidelity gains are actually about better explanations, rather than just larger shifts.

---

> ### Author Response · Authors · 2024-11-20
> **Rebuttal by Authors**
>
> Thanks for your review!
>
> Our responses to your concerns are as follows:
>
> **Local Fidelity:**
> We understand your concerns regarding feature-level local fidelity. In fact, high local fidelity is a key property of a high-quality local explanation. However, we'd like to argue that it's worth nothing to only stick to feature-level local fidelity.
> Concept-based explanations provide concept-level local fidelity. As concept-based explanations capture model behavior more faithfully and are more intuitive, they offer a more natural and convenient way for end-users to understand and utilize the explanations (Sun et al., 2023[1]). In other words, users are more inclined to know how the models behave in the concept-level locality.
>
> **Quality of Concepts:**
> We are unclear about how to use "concept relevance" and "coherence" to measure concept quality. To our knowledge, TCAV is a method for attributing importance to concepts, similar to what ConLUX-augmented LIME and KernelSHAP do. We'd appreciate it if you can explain it in detail. We would be happy to address your concerns further.
> Moreover, in both our text and image experiments, the quality of discovered concepts is proved to be human-understandable. We extract concepts by workflows similar to those used in previous methods: TBM (Ludan et al., 2023[2]) for text and EAC (Sun et al., 2023[1]) for images. These prior works have validated that the generated concepts are understandable through human evaluations.
>
> **Fairness and Reliability:**
> We acknowledge that the capabilities of pre-trained models can influence the quality of explanations. However, this should not be considered a disadvantage of ConLUX. On one hand, our experiments demonstrate that ConLUX achieves state-of-the-art performance. On the other hand, as large models continue to advance, their fairness and reliability will improve, enabling ConLUX to generate explanations of higher quality.
>
> **Scale of perturbation:**
> The improvement in fidelity metrics is not brought by a larger perturbation scale. Considering the experiment of image classification in Section 4, ConLUX improves the fidelity metrics, while for both vanilla and ConLUX-augmented methods, the perturbation scale is the same, i.e. from the original image to a fully masked image.
>
>
> [1] Ao Sun, Pingchuan Ma, Yuanyuan Yuan, and Shuai Wang. Explain any concept: Segment anything meets concept-based explanation. (arXiv:2305.10289), May 2023. doi: 10.48550/arXiv.2305.10289. URL http://arxiv.org/abs/2305.10289. arXiv:2305.10289 [cs].
>
> [2] Josh Magnus Ludan, Qing Lyu, Yue Yang, Liam Dugan, Mark Yatskar, and Chris CallisonBurch. Interpretable-by-design text classification with iteratively generated concept bottleneck. (arXiv:2310.19660), October 2023. doi: 10.48550/arXiv.2310.19660. URL http://arxiv.org/abs/2310.19660. arXiv:2310.19660 [cs].

---

> > ### Comment · Reviewer_CduZ · 2024-11-23
> >
> > Thank you for your response.
> >
> > Regarding local fidelity, I appreciate your point about concept-level fidelity being more intuitive for end-users. However, my concern lies in the potentially drastic nature of the shift introduced by concept-level perturbations compared to feature-level adjustments. This shift could risk disrupting the local neighborhood of the input, which is critical for maintaining fidelity in explanations. Unlike feature-level perturbations, which make smaller adjustments (e.g., changing a single word or pixel), concept-level perturbations often involve broader, higher-order changes that may inadvertently alter the input more significantly.  I recommend running controlled experiments that measure fidelity across varying perturbation scales. Additionally, the broader nature of concept-level perturbations could inadvertently inflate recorded responses in metrics like AOPC or coverage.
> >
> > Regarding concept quality, while I understand your concerns about directly using TCAV, the broader idea is to introduce a mechanism to evaluate the relevance and coherence of the concepts extracted by ConLUX. This would help substantiate your claim that these concepts align well with the model's decision-making process across different domains.
> >
> > For example if used TCAV:
> >
> > - Relevance: TCAV's directional derivative approach could be used to assess how strongly the extracted concepts influence the target model's output. By checking whether the concepts identified by ConLUX align with the decision boundary of the model, you could measure how relevant these concepts are in the context of specific predictions.
> > - Coherence: TCAV can also reveal whether the concepts are consistently meaningful across instances within a domain. If certain concepts show a consistent, high relevance score across inputs, this could serve as evidence of their coherence and general applicability.

---

> > > ### Author Response · Authors · 2024-12-03
> > >
> > > Thanks again for your feedback.
> > >
> > > As mentioned earlier, TCAV is an attribution method similar to LIME, KSHAP, DeepLift, and others. While we believe the fidelity results sufficiently demonstrate the high quality of our concept-level predicates, we are happy to provide further clarification through the attribution results.
> > >
> > > We compared the highest attributed predicates in LIME, KSHAP, and their ConLUX-augmented explanations. The results show that, in our text experiments, on average, the predicate with the highest attribution score in ConLUX-augmented explanations receives more than 3 times the attribution score compared to those in the feature-level explanations. In our image experiments, on average, the highest attributed concept predicates receive 1.5 times the score of the highest attributed feature predicates.

---

> ### Author Response · Authors · 2024-11-24
>
> Regarding your concerns, our responses are as follows:
>
> **Local fidelity**
> We would like to argue that not all types of explanation methods should guarantee feature-level local fidelity. For concept-based explanations, concept-level local fidelity is the key metric. In fact, any explanation technique that uses predicates and perturbations above the feature level inherently risks a loss in feature-level fidelity.
> Essentially speaking, the primary goal of using a metric of explanation is to demonstrate that an explanation is useful for its target users. As we mentioned before, end-users are shown to be more inclined to know how the models behave in the concept-level locality. Specifically, they care about how the model responds to changes in concepts rather than individual features. Therefore, we focus on measuring concept-level fidelity for concept-based explanations, as is standard practice in prior works[1-4]. Since end-users do not prioritize output changes caused by feature-level perturbations, a higher feature-level fidelity does not make concept-based explanations more useful. As such, measuring feature-level local fidelity for concept-based explanations is unnecessary.
>
> **Fidelity Results & Perturbation Scale:**
> We don't think our fidelity results are inflated.
> - AOPC: In Figure 5, the values along the x-axis represent K%, which indicates the proportion of changed predicates rather than the number of changed predicates. This provides a fair comparison. If you believe there is an issue with this metric, could you please provide a specific example to clarify your concern?
> - Coverage: ConLUX improves both precision and coverage, meaning our explanations better predict the model's behavior across more data and with greater accuracy. This makes our explanations demonstrably superior to the vanilla methods. Could you please explain in detail why you think our explanations might be potentially worse?
>
> **TCAV**
> Directional derivative is just one method for attributing importance to features or concepts, similar to other techniques like LIME, LRP, or DeepLift. As noted in prior works, these methods are not always faithful [5], so directional derivative should not be used as a metric to evaluate the concepts.
>
> **Quality of Concepts**
> We think the alignment between our concepts and the target models has already been evaluated through our fidelity experiments. Specifically, higher fidelity in a ConLUX-augmented explanation as a whole indicates that the combination of multiple concepts forming the explanation aligns better with the target model. Additionally, the AOPC curve demonstrates that, for any proportion \( K\% \), the highest-attributed concepts align better with the target model.
>
> Regarding the two metrics you introduced:
>
> - **Concept Relevance:** This metric appears to measure the importance of individual concepts. However, we go beyond mere attribution by also evaluating the quality of this attribution through fidelity experiments.
> - **Concept Coherence:** We are unclear on the necessity of this metric. For end-users who aim to understand the local behavior of target models or predict their outputs in a specific locality, we didn't notice any practical difference between two local explanations with the same fidelity and understandability but differing in Concept Coherence. Could you please explain why Concept Coherence is critical in this context?
>
>
>
> [1]Fel, Thomas, Victor Boutin, Louis Béthune, Remi Cadene, Mazda Moayeri, Léo Andéol, Mathieu Chalvidal, and Thomas Serre. “A Holistic Approach to Unifying Automatic Concept Extraction and Concept Importance Estimation.” Advances in Neural Information Processing Systems 36 (December 15, 2023): 54805–18.
>
> [2]Sun, Ao, Pingchuan Ma, Yuanyuan Yuan, and Shuai Wang. “Explain Any Concept: Segment Anything Meets Concept-Based Explanation.” arXiv, May 17, 2023. https://doi.org/10.48550/arXiv.2305.10289.
>
> [3]Fel, Thomas, Agustin Picard, Louis Bethune, Thibaut Boissin, David Vigouroux, Julien Colin, Rémi Cadène, and Thomas Serre. “CRAFT: Concept Recursive Activation FacTorization for Explainability.” arXiv, March 28, 2023. https://doi.org/10.48550/arXiv.2211.10154.
>
> [4]Zaval, Mounes, and Sedat Ozer. “Improving the Explain-Any-Concept by Introducing Nonlinearity to the Trainable Surrogate Model.” arXiv, June 24, 2024. https://doi.org/10.48550/arXiv.2405.11837.
>
> [5]Sundararajan, Mukund, Ankur Taly, and Qiqi Yan. “Axiomatic Attribution for Deep Networks.” arXiv, June 12, 2017. https://doi.org/10.48550/arXiv.1703.01365.

---

### Official Review · Reviewer_SUB7 · 2024-11-04

**Soundness:** 2
**Presentation:** 2
**Contribution:** 2
**Rating:** 5
**Confidence:** 3

**Summary:**

The paper proposes to use foundation models to discover concepts to augment on methods like LIME to provide concept-based local explanations. The method is evaluated on sentiment analysis and image classification tasks.

**Strengths:**

The proposed framework is interesting and can be useful if validated rigorously. The paper explores two different modalities. The framework is applied across multiple established methods (LIME, Kernel SHAP, Anchor, and LORE).

**Weaknesses:**

Concept discovery using LLM is the key aspect of the proposed framework. This calls for a human evaluation to answer the question: are the concepts discovered by the foundation models indeed aligned with a human-understandable representation? Currently, it sounds like it is assumed that the prompt will take care of this.


How faithful is the backtracking from the perturbed concept space to the original space? This also done by the LLM?

Methods like LIME create a local approximation of the original function to a human-understandable form and explain the decision. This is the reliable yet explainable part of the method. By using LLM for discovering concepts, (given each task and the sample), it becomes difficult even locally to explain the predicates. The decision-making is not fully explained, i.e., the choice of predicates cannot be explained. For instance, in case a poor predicate is chosen by an LLM, the user may be confused by the explanation. I think the proposed method, by using foundation models for concept discovery, makes the framework less reliably explainable.

To prove the robustness of the method, it would be useful to try intervention-based causal metrics, especially since the concept discovery is done by foundation models. C-insertion and deletion are often used to evaluate concept importance and fidelity in concept-based explanations.

Robustness to the prompt: how robust is the framework to the construction of the prompt? Also, given the literature around prompt engineering, the framework could explore what might be the optimal prompting strategy for discovering human-interpretable concepts.


The paper proposes a new framework for explanation where concept discovery is done by foundation models. For the text modality, the experiments are restricted to sentiment analysis, and 1000 images from the ImageNet dataset are used for classification for images. The method calls for more experiments for validation. However, this is not the sole reason for the decision.


There is no baseline comparison. Though the method is a new paradigm, it could be compared with concept bottleneck methods or predicates selected by other methods. There could also be an ablation study on different LLM model combinations. Additionally, the authors could discuss concept bottleneck-based methods. What do you think about a method where task-specific concept bottlenecks are chosen by the LLM? This is not a weakness or a mandatory experiment for the text.


Minor comments:

L44: Please provide a citation, as not all visual explanation methods using attribution mapping follow this principle.
L80: "Moreover, we observe that ... across different tasks." given the restricted number of tasks evaluated it might be a good idea to tone down this claim.
Teaser figure: I think, this can be made more comparable if the attribution method is shown with certain thresholding. When shown post-thresholding, the viz can clearly demonstrate the benefit of the proposed method, even allowing for certain comparative evaluations later.
L93: This is a tradeoff between reliability of explanation to prediction and understandability. If the model is indeed making a decision based on the token, pivoting the explanation on different concepts can change it.
Figure 2: Comparison of LIME and ConLUX-augmented LIME; the ConLUX highlights the kid! Is this a good explanation? Of course, SAM performs a good segmentation of the image, providing an object-level explanation, but the attribution seems incorrect to me. Or did I miss something?

**Questions:**

For each data point, the concept used can be different. Given that it comes from an LLM, there can even be stochasticity over it (it may also be worth mentioning the LLM temperature setting for this, if not already mentioned). Even otherwise, two very similar reviews can have different concepts chosen for explanation, with no theoretical guarantee on the LLM choosing similar predicates. This lack of control makes the explanation less robust. In theory, changing a token (may be an adversarially crafted one in practice) can change the entire explanation, as the choice of predicates is left to the LLM. One cat may be identified by its ears and a similar one by its eyes. Can you propose an experiment to study this robustness?

---

> ### Author Response · Authors · 2024-11-20
> **Rebuttal by Authors**
>
> Thanks for your review!
>
> First of all, we would like to clarify that we propose the framework ConLUX to automatically extract high-level concepts and incorporate them into existing local model-agnostic explanation techniques to provide concept-based explanations. To achieve this, we make the following contributions:
> - We introduce a unified approach to augment various existing local model-agnostic explanation techniques to provide concept-based explanations. This requires the ability to automatically extract concepts and perform bidirectional mapping between concept level and feature level.
> - We find that large pre-trained models can be used to extract concepts and perform bidirectional mapping between concept level and feature level. Specifically, existing works (Ludan et al., 2023[1]; Sun et al., 2023[2]) have utilized large models to extract concepts and conduct feature-to-concept mapping (i.e., determining whether an instance satisfies a specific concept) for specific tasks. We generalize these findings and show that large pre-trained models can also perform concept-to-feature mapping (i.e., generating samples based on changes in concept-level information), thus enabling the entire workflow to be executed.
>
> We validate the feasibility of ConLUX by two instantiations for image and text data. The results show that the current instantiations of ConLUX achieve state-of-the-art performance.
> Our focus is not on identifying the best pre-trained models or prompts. Instead, ConLUX is designed to be instantiated with any large pre-trained models capable of performing bi-directional mappings. As large models and prompts continue to advance, the explanations provided by ConLUX will naturally improve alongside these developments.
>
> Our responses to your specific concerns are as follows:
>
> **Alignment of concepts and human-understandable representations:**
> In both our text and image experiments, the discovered concepts are proved to be human-understandable. We extract concepts by workflows similar to those used in previous methods: TBM (Ludan et al., 2023[1]) for text and EAC (Sun et al., 2023[2]) for images. These prior works have validated that the generated concepts are understandable through human evaluations.
>
> **Concept level perturbation:**
> For image data, we perform concept-level perturbation by masking, which is naturally faithful. For text data, we propose to perform concept-level perturbation by an LLM, and we admit it's worth verifying the fidelity of concept-level perturbation. As Ludan et al. (2023)[1] have shown that LLMs can check whether an instance satisfies a given concept, we are conducting an experiment to evaluate the consistency of LLM-based perturbations. Specifically, we use LLMs to verify whether the perturbation alters the concept as intended. We will report these results in a subsequent update.
>
> **Explanation of the choice of predicates:**
> We would like to argue that it is unnecessary for an XAI technique to be self-explanable. While it is possible that a large pre-trained model may select suboptimal predicates, Sun et al. (2023)[2] have demonstrated through human evaluations that, on average, the concepts discovered by pre-trained models are more understandable to users than feature-level predicates.
>
>
> **C-insertion and Deletion:**
> We provide the results of the deletion experiments in Figure 5 and Table 2.
>
> **Robustness to the prompt:**
> The robustness to prompts is a property of the pre-trained models themselves rather than our framework. Our work does not aim to improve the large pre-trained models or prompts. Instead, as large models and prompts continue to advance, the explanations generated by our framework will also naturally improve.
>
> **More experiments:**
> We are conducting an experiment on a text-generation task and incorporating additional images into our image classification task. We will report these results in a subsequent update.
>
> **Baseline comparison:**
> In our paper, we compare ConLUX with TBM, a bottleneck model for the sentiment analysis task. Specifically, our framework focuses on generating concept-level explanations locally, without requiring internal information about the target models. We compare our framework with two state-of-the-art task-specific techniques for providing explanations for black-box target models: TBM and EAC. Please refer to Section 4 and Table 3 for further details.
>
> **L44:**
> We apologize that we may not fully understand the issue you mentioned. We'd appreciate it if you can explain more. Additionally, we have added a citation in our edited version.

---

> ### Author Response · Authors · 2024-11-20
> **Rebuttal by Authors (Continued)**
>
> **L80:**
> Theoretically, given that large pre-trained models are capable of obtaining concepts for various types of input data, such as natural language, images, and medical data (Chen et al., 2024[3]), and that existing model-agnostic techniques are applicable across a wide range of tasks, including classification and generation (Paes et al., 2024[4]), we make the claim that large pre-trained models can extract high-level concepts across different tasks.
> As it is impractical to enumerate all possible tasks, we chose two popular tasks to instantiate ConLUX in our experiments. However, as mentioned before, we are happy to include an experiment on a text-generation task to further support this claim.
>
>
> **Teaser figure with thresholding:**
> Thanks for your suggestion. However, we have a different perspective. Deciding which threshold to use in visualizations is a non-trivial task and is not inherently part of attribution techniques like LIME or KernelSHAP. Therefore, we believe it is better to present our visualizations without including a threshold.
>
> **L93:**:
> This example is not a tradeoff between the reliability of explanations and their understandability. Elevating explanations from the feature level to the concept level enhances both fidelity and understandability.
> If the model's decision-making is indeed based solely on individual tokens, such a tradeoff can exist. However, this issue is a common limitation of concept-based explanations. Moreover, target models with such behavior are typically very simple (e.g., linear models), which are not sufficiently powerful and are rarely used in practice. Additionally, such simple models are often inherently self-explanable. Consequently, this issue is unlikely to pose a significant obstacle to applying concept-based methods.
>
>
> **Figure 2:**
> ConLUX highlighting the kid indicates the limitation of YOLOv8, which takes the kid into consideration when identifying a punching bag.
>
> **Response to the question:**
> We use the default temperature setting for each LLM, which is 0 for GPT-3.5, and 0.8 for Llama 3.1.
>
> > Two very similar reviews can have different concepts chosen for explanation......One cat may be identified by its ears and a similar one by its eyes. Can you propose an experiment to study this robustness?
>
> We'd like to clarify this issue from the following two perspectives:
>
> - **In-distribution data**:
>   For the instances in the test set of the used dataset, our experiments demonstrate that ConLUX explanations are faithful, and previous works (Ludan et al., 2023[1]; Sun et al., 2023[2]) demonstrate that high-level concepts generated by these large pre-trained models are understandable. This indicates that ConLUX performs well for in-distribution data. In this case, if two similar instances are assigned different concepts and explanations, these explanations can still be both faithful and understandable. This can be viewed as explaining the same decision from two different perspectives. Developing a method to provide explanations from multiple perspectives is an interesting direction for future research.
>
> - **Out-of-distribution (OOD) adversarial samples**:
>   Suppose the two similar instances include one in-distribution instance and one out-of-distribution (OOD) adversarial sample. In that case, it is possible for concept-based explanations to be unfaithful on the adversarial sample. However, we do not consider this a weakness of our method.
>   On the one hand, the potential low fidelity is a common risk of concept-based explanations:
>   1) For either large models or other methods used to discover concepts (e.g., extracting concepts from gradients, activations, or attention weights), there is a risk of generating poor-quality concepts for adversarial instances on out-of-distribution (OOD) adversarial samples.
>   2) The fidelity of explanations cannot be guaranteed in such cases, as the decision-making process for adversarial samples may not align with the extracted concepts.
>
>    On the other hand, concept-based explanations are intended for non-expert end-users. Explaining adversarial samples is beyond the scope of this design. For debugging adversarial samples, expert users should utilize explanation techniques specifically designed for expert use cases.
>
> [1] Ludan, J. M., et al. (2023). Interpretable-by-design text classification with iteratively generated concept bottleneck. *arXiv preprint*, arXiv:2310.19660.
>
> [2] Sun, A., et al. (2023). Explain any concept: Segment anything meets concept-based explanation. *arXiv preprint*, arXiv:2305.10289.
>
> [3] Chen, Y., et al. (2024). BURExtract-Llama: An LLM for clinical concept extraction in breast ultrasound reports. *Proceedings of MCHM’24*, 53–58.
>
> [4] Paes, L. M., et al. (2024). Multi-level explanations for generative language models. *arXiv preprint*, arXiv:2403.14459.

---

> ### Author Response · Authors · 2024-12-03
>
> - We have conducted the experiment to validate the faithfulness of LLM perturbation in our text experiments. Specifically, we test whether Llama-3.1, the LLM used in our experiments, can generate sentences with the expected concepts. The results show that Llama-3.1 generates sentences as expected with over 99% accuracy.
>
> - We have performed an experiment on a text-generation task. Specifically, we explain the Llama-3.1 model on a text summarization task. We randomly selected 20 sentences from the CNN/Daily Mail dataset, and prompted Llama-3.1 to perform text summarization. Paes et al. (2024) introduce C-LIME and L-SHAP to generate attribution explanations for generative tasks. We augmented these two methods with ConLUX and compared the fidelity of the vanilla explanations with the ConLUX-augmented ones. The average AOPC values are shown below:
>
>
> | C-LIME | LIME* | L-SHAP | KSHAP* |
> |--------|-------|--------|--------|
> | 0.181  | **0.243** | 0.171  | **0.256**  |
>
>
> These results show that ConLUX can also improve the fidelity of explanations for a text-generation task.

---

> ### Author Response · Authors · 2024-12-04
>
> We run our image experiments on 5000 more images, and the final results are as follows:
>
>
> **Coverage(\%)**
>
> |           | Anchors | Anchors* | LORE | LORE* |
> |-------|--------|-------|--------|--------|
> |YOLOv8| 28.9  | **31.1** | 21.4  | **24.9**  |
> |ViT|25.3| **28.7**|20.9|**23.8**|
> |ResNet-50|27.2| **31.0** | 20.2 |**29.8**|
>
> **Precision(\%)**
> |           | Anchors | Anchors* | LORE | LORE* |
> |-------|--------|-------|--------|--------|
> |YOLOv8| 85.2  | **99.0** | 86.4 | **92.0**  |
> |ViT|89.1|**98.6**|90.1|**95.9**|
> |ResNet-50| 90.5|**99.1**|84.9|**92.4**|
>
> **AOPC**
>
> |           | LIME | LIME* | SHAP | KSHAP* |
> |-------|--------|-------|--------|--------|
> |YOLOv8| 0.411  | **0.498** | 0.457  | **0.603**  |
> |ViT|0.501 | **0.643**|0.489|**0.635**|
> |ResNet-50|0.234| **0.317**|0.243|**0.335**|
>
> **Accuracy_a**
> |           | LIME | LIME* | SHAP | KSHAP* |
> |-------|--------|-------|--------|--------|
> |YOLOv8| 0.137  | **0.050** | 0.328  | **0.051**  |
> |ViT|0.226|**0.055**|0.246|**0.078**|
> |ResNet-50| 0.359|**0.126**|0.314|**0.079**|

---

### Author Response · Authors · 2024-12-01
**Response to All Reviewers**

Thanks to all the reviewers for your thoughtful feedback. We have uploaded a revised version of the paper to address the main concerns raised:

1. **Clarification of Our Contribution (Paragraph starting at line 79):**
   - We introduce a unified approach to augment various existing local model-agnostic explanation techniques to provide concept-based explanations. This requires the ability to automatically extract concepts and perform a bidirectional mapping between the concept level and the feature level.
   - We show that large pre-trained models can be used to extract concepts and perform bidirectional mapping between the concept level and feature level. Specifically, we find existing works (Ludan et al., 2023[1]; Sun et al., 2023[2]) have utilized large models to extract concepts and conduct feature-to-concept mapping (i.e., determining whether an instance satisfies a specific concept) for specific tasks. We generalize these findings and show that large pre-trained models can also perform concept-to-feature mapping (i.e., generating samples based on changes in concept-level information), thereby enabling the full workflow.

2. **Validation of Concept-Level Perturbation (Line 340):**
   In our text experiments, we use LLMs to perform the perturbation. We have conducted an experiment to validate the faithfulness of this perturbation. Specifically, we test whether Llama-3.1, the LLM used in our experiments, can generate sentences with the expected concepts. The results show that Llama-3.1 generates sentences as expected with over 99% accuracy.

You can refer to our responses below each review for other questions and concerns. We'll appreciate your further feedback.

---

### Meta-Review · Area_Chair_JaPL · 2024-12-20

**Metareview:**

The paper introduces an intriguing method using large pre-trained models for concept discovery and perturbation, but several issues limit its reliability and generalizability. Concept-level perturbations can distort inputs, failing to capture local decision boundaries effectively, especially when modeled as entire objects. The extracted concepts lack evidence of generalization across domains, raising concerns about interpretability. Reliance on pre-trained models introduces biases, with no mitigation strategies proposed. Additionally, LLM-based perturbations in language tasks face fidelity issues, and the framework's sensitivity to prompt selection impacts reproducibility. Significant revisions, including finer-grained experiments, robust validation, and bias mitigation, are needed to improve its rigor and reliability.

**Additional Comments On Reviewer Discussion:**

Key weakness raised by reviewers:
1. Concept Alignment: Lack of explicit human evaluation to confirm discovered concepts align with human-understandable representations.
2. Fidelity of Perturbations: Concerns about the reliability of LLM-based perturbations, especially for text data.
3. Predicate Selection: LLM-selected predicates may be inconsistent or confusing, impacting explanation quality.
4. Prompt Robustness: Framework is sensitive to prompt design, with no exploration of optimal strategies.
5. Limited Experiments: Experiments are restricted to specific tasks (sentiment analysis, ImageNet) with insufficient validation.
6. Baseline Comparisons: No thorough comparison with concept bottleneck methods or ablation studies.
7. Broad Perturbations:Concept-level changes may distort inputs significantly, risking local fidelity.

Author Responses:
1. Highlighted prior methods validating concept understandability and ongoing validation experiments.
2. Preliminary results show high fidelity (>99%) for LLM-based perturbations.
3. Argued that self-explanation of predicates is unnecessary and cited prior studies supporting concept understandability.
4. Claimed prompt robustness is tied to pre-trained models and will improve as models advance.
5. Ongoing experiments aim to address limited validation and expand tasks.
6. Justified existing comparisons with state-of-the-art techniques as appropriate for the framework.
7. Defended broader perturbations as improving concept-level fidelity and user comprehension.

While the authors address concerns with explanations and ongoing experiments, some key weaknesses, such as robustness and validation, remain unresolved, relying on future work.

---

### Decision · Program_Chairs · 2025-01-22

Reject